# Are Young People with Turner Syndrome Who Have Undergone Treatment with Growth and Sex Hormones at Higher Risk of Metabolic Syndrome and Its Complications?

**DOI:** 10.3390/biomedicines12051034

**Published:** 2024-05-08

**Authors:** Mariola Krzyścin, Elżbieta Sowińska-Przepiera, Karolina Gruca-Stryjak, Ewelina Soszka-Przepiera, Igor Syrenicz, Adam Przepiera, Žana Bumbulienė, Anhelli Syrenicz

**Affiliations:** 1Pediatric and Adolescent Gynecology Clinic, Department of Gynecology, Endocrinology and Gynecological Oncology, Pomeranian Medical University in Szczecin, Unii Lubelskiej 1, 71-252 Szczecin, Poland; 2Department of Endocrinology, Metabolic and Internal Diseases, Pomeranian Medical University in Szczecin, Unii Lubelskiej 1, 71-252 Szczecin, Poland; 3Department of Perinatology and Gynecology, Poznan University of Medical Sciences, 60-535 Poznań, Poland; 4Centers for Medical Genetics GENESIS, ul. Dąbrowskiego 77a, 60-529 Poznań, Poland; 5II-nd Department of Ophthalmology, Pomeranian Medical University in Szczecin, Al. Powstancow Wielkopolskich 72, 70-111 Szczecin, Poland; 6Department of Urology and Urologic Oncology, Pomeranian Medical University in Szczecin, Al. Powstańców Wielkopolskich 72, 70-111 Szczecin, Poland; 7Center for Obstetrics and Gynecology, Vilnius University Hospital Santaros Klinikos, Vilnius University, Faculty of Medicine, LT-03101 Vilnius, Lithuania

**Keywords:** Turner syndrome, metabolic syndrome, body fat, visceral fat, recombinant human growth hormone, hormone replacement therapy, monosomy, mosaic karyotype

## Abstract

Introduction: Metabolic syndrome (MetS), characterized by visceral obesity, glucose abnormalities, hypertension and dyslipidemia, poses a significant risk of diabetes and cardiovascular disease. Turner syndrome (TS), resulting from X chromosome abnormalities, carries health complications. Despite growing evidence of an increased risk of MetS in women with TS, its prevalence and risk factors remain under investigation. These considerations are further complicated by the varying timing and dosages of treatment with growth hormone and sex hormones. Methods: We conducted a cross-sectional study comparing 44 individuals with TS with 52 age-matched control subjects. Growth hormone treatment in the study group was administered for varying lengths of time, depending on clinical response. We collected anthropometric, metabolic, endocrine and body composition data. Statistical analyses included logistic regression. Results: Baseline characteristics, including age, BMI and height, were comparable between the TS and control groups. Hormonally, individuals with TS showed lower levels of testosterone, DHEA-S, and cortisol, as well as elevated FSH. Lipid profiles indicated an atherogenic profile, and the body composition analysis showed increased visceral adipose tissue in those with TS. Other metabolic abnormalities were common in individuals with TS too, including hypertension and impaired fasting glucose levels. The risk of MetS components was assessed in subgroups according to karyotypes: monosomy 45X0 vs. other mosaic karyotypes. Logistic regression analysis showed a significant association between increased visceral adipose tissue in subjects with TS. Those with metabolic complications tended to have less muscle strength compared to those without these complications in both the study and control groups. Conclusions: This study highlights the unique metabolic and cardiovascular risk profile of individuals with TS, characterized by atherogenic lipids, higher levels of visceral adipose tissue and increased metabolic abnormalities. These findings underscore the importance of monitoring metabolic health in individuals with TS, regardless of age, BMI or karyotype, and suggest the potential benefits of lifestyle modification, building more muscle strength, and weight control strategies. Further research is needed to better understand and address the metabolic challenges faced by women with TS.

## 1. Introduction

Metabolic syndrome (MetS), a multifaceted pathological condition characterized by a combination of visceral obesity, abnormalities in glucose metabolism, hypertension, and dyslipidemia, poses a significant public health burden due to its heightened association with the development of diabetes and cardiovascular disease [1,2,3]. In parallel, Turner syndrome (TS), arising from the complete or partial absence of the X chromosome and affecting one in 2000–2500 live female births, is commonly linked to delayed growth, reduced final height, and gonadal dysgenesis, thereby increasing the risk of various health complications [4,5,6].

While there is an expanding body of evidence indicating an elevated relative risk of diabetes, ischemic heart disease, atherosclerosis, hypertension, and obesity in individuals with TS [7,8,9,10,11], the specific prevalence and contributing risk factors for MetS in this population remain subjects of ongoing investigation. Moreover, the utilization of recombinant human growth hormone (rhGH) and hormone replacement therapy (HRT) in young females with TS adds complexity to predicting their effects on metabolic status [12,13,14]. It is noteworthy to mention that the indications for rhGH in patients with TS are nuanced, as this group is not a straightforward growth hormone deficient (GHD) population [15,16]. Consequently, rhGH therapy may not yield the same degree of benefits observed in patients with GHD. This aspect introduces an additional layer of complexity to the therapeutic landscape for individuals with TS [17].

The present study is based on the hypothesis that effective hormonal therapies can affect the prevalence of MetS parameters in individuals with TS [13,14]. The main objective of this study is to evaluate the prevalence of MetS elements in young women with TS who have previously undergone successful rhGH treatment and are currently receiving sequential estrogen–progesterone therapy. We aimed to investigate whether achieving the “clinical effect” of growth hormone treatment, assumed as a minimum of 150 cm of ultimate height in women with TS, will bring their metabolic profile and body composition closer to that of healthy women. Additionally, concentrations of metabolic and hormonal parameters, with a focus on those related to diabetes mellitus, will be measured. Body composition will also be analyzed in these patients, and the results will be comparatively assessed against an age- and anthropometrically matched control group.

## 2. Materials and Methods

### 2.1. Subjects

We conducted the study between January 2021 and February 2022 at a single referral center of a university hospital. It included young female patients between the ages of 18 and 28. It included 44 patients with TS and 52 women in the control group. The inclusion criteria were the presence of genetic abnormalities warranting a diagnosis of TS and a minimum height of 150 cm. The exclusion criteria for the study were a height of less than 150 cm or more than 162 cm. There were no patients with type 1 diabetes in the study group, nor any with major cardiovascular or renal defects. One patient suffered from celiac disease, and eight women from autoimmune thyroid disease. In conclusion, the patients in the study group did not use any chronic treatment except levothyroxine. In the questionnaire, only two female patients with TS declared occasional cigarette smoking (less than once a month), and none of them admitted to alcohol abuse.

All patients with TS had stopped growing, and their karyotypes consisted of monosomy X (*n* = 21), classic 45.X/46.XX mosaicism (*n* = 16) or other double or triple mosaics with at least two abnormal cell lines (*n* = 7). Every girl diagnosed with TS received rhGH treatment at a customized dosage (1.2–2 mg/m^2^/day) based on their clinical response. The average duration of growth hormone (GH) therapy was 5.0 years, spanning an age range from 1.3 to 12.5 years. Estrogen–progesterone therapy was used in 37 patients to initiate sexual maturation and in 3 patients for secondary ovarian failure, while 4 patients showed signs of spontaneous sexual maturation. Transdermal sequential estrogen–progesterone therapy was initiated at a mean age of 12.1 years (age range 10.9–13.5 years), with a starting dose of 17β-estradiol of 0.00625 mg/day, gradually increased to 0.05 mg/day. After an average of 2.4 years, transdermal therapy was continued in a sequential regimen with the same dose of 17β-estradiol for 2 weeks and 17β-estradiol plus norethisterone acetate for another 2 weeks at a dose of 0.17 mg/day. Importantly, subjects from the control group did not receive rhGH treatment, ensuring a clear distinction between the treatment and control groups in evaluating the effects of growth hormone therapy on various parameters.

The control group (CG) constituted of 52 age-, height- and weight-matched, healthy, premenopausal women, who were recruited in the Outpatient Clinic of Endocrinology, Pomeranian Medical University in Szczecin, Poland. All subjects in the CG were Caucasian, were equal to or taller than 150 cm but no taller than 162 cm, had regular menstrual cycles, had not undergone estrogen, progesterone or testosterone therapy, did not take any contraceptives, did not smoke or abuse alcohol, and had not given birth to any child in the past. The exclusion criteria for the study based on height (less than 150 cm or more than 162 cm) were implemented to ensure homogeneity within the study groups, which is particularly relevant given its association with the diagnosis and treatment of TS. It aimed to minimize potential confounding factors related to height variations.

All patients underwent standard history and physical examinations. Anthropometric measurements were taken (height was measured in a standing position, using a digital telescopic wall-mounted stadiometer and weight was determined to the nearest 0.1 kg using an electronic scale), on the basis of which the body mass index was calculated. Waist circumference (WC) was obtained at the midpoint between the lowest rib and the iliac crest.

Systolic and diastolic blood pressures (SBP and DBP) were measured using an automated sphygmomanometer monitor. Following 10 min of rest, three readings in the sitting position were obtained from the left arm with an appropriate size cuff at 5 min intervals and were then averaged.

### 2.2. Biochemical Analyses

Fasting blood samples were collected from all subjects during the follicular phase of the menstrual cycle, specifically on the 3rd, 4th, or 5th day of the menstrual cycle. These samples underwent testing in the hospital’s prospective laboratory using standard assays, including Luminex, Beckman DXC 800, GM7 analyzer, RIA, and the Human IGF-I Quantikine ELISA Kit from R&D Systems, located at the study’s core laboratory of Pomeranian Medical University in Szczecin.

### 2.3. Mucsle Function Assessment

Handgrip strength was measured using a SAEHAN 5030J1 hydraulic manual dynamometer with a 40 kg load. The test assessed isometric hand and forearm strength. Patients stood during the assessment, with their tested arm bent at a 90 degree angle and in contact with their body. Each patient was instructed to squeeze the dynamometer bar three times with maximum effort before resting for 30 s, first with their right hand and then with their left hand. The average of these three measurements was recorded as the final muscle strength measurement, expressed in kilograms.

### 2.4. Assessment of Body Composition

To assess bone mineralization, a body composition analysis of the study participants was conducted using the DXA method. This analysis employed a GE Lunar Prodigy instrument from Madison, WI, USA, equipped with CoreScan ™ H8801CP automatic software and an automatic whole-body scanning method utilizing the manufacturer’s original software for body composition. The parameters assessed included fat mass (BF) and visceral fat mass (VF).

### 2.5. Diagnosis of Metabolic Syndrome

Participants met the criteria for a diagnosis of MetS if they had at least three of the following: abdominal obesity—waist circumference (WC) ≥ 80 cm, hypertriglyceridemia (≥150 g/dL), low HDL-C < 50 mg/dL, high blood pressure (systolic blood pressure ≥ 130 mmHg or diastolic blood pressure ≥ 85 mmHg) or mean arterial pressure (MAP) ≥ 100 mmHg, and high fasting glucose (≥100 mg/dL). The definition of MetS was based on the updated harmonized criteria from the International Diabetes Federation (IDF).

### 2.6. Statistical Analysis

The statistical analysis aimed to explore the associations between various factors and TS by employing multiple logistic regression. Descriptive statistics, such as medians and standard deviations (SD), were used to present variable characteristics. ANOVA or Kruskal–Wallis tests were employed, based on meeting the ANOVA assumptions for comparing parameters between two groups (with or without TS).

Backward step-wise multivariate logistic regression was conducted with a subsequent step-wise elimination process based on AIC and *p*-value criteria (with a cutoff of 0.05) for variable selection. Interactions were added and removed if found to be non-significant. The variables considered in the analysis included age, body mass index (BMI), systolic blood pressure (SBP), diastolic blood pressure (DBP), fasting glucose level (FGL), insulin, homeostatic model assessment for insulin resistance (HOMA-IR), total cholesterol (TC), high-density lipoprotein (HDL), low-density lipoprotein (LDL), triglyceride (TG), total calcium (Ca), ionized Ca, thyroid-stimulating hormone (TSH), free thyroxine (FT4), follicle-stimulating hormone (FSH), estradiol, prolactin, testosterone, dehydroepiandrosterone sulfate (DHEAS), cortisol, adrenocorticotropic hormone (ACTH), insulin-like growth factor 1 (IGF-1), vitamin D, body fat (BF), and visceral fat (VF). The dependent variable was categorized into two groups: individuals with TS (TS group) and height- and weight-matched controls (CG).

All statistical analyses were performed using the R statistical platform (version 4.2.3, 2023; RRID:SCR_001905, https://cran.r-project.org, accessed on 22 February 2024).

## 3. Results

The demographic and anthropometric characteristics presented in Table 1 reveal that, for the variables examined (age, BMI, height, and weight), there are no statistically significant differences between individuals with TS and the control group (CG). These findings suggest that any observed differences in the subsequent analyses are less likely to be influenced by variations in these baseline characteristics, making the comparison between the two groups more reliable and informative. The difference in WC between the groups was not statistically significant.

Table 2 presents clinical and laboratory characteristics comparing individuals with TS to the controls. There are no statistically significant differences in SBP, DBP, metabolic parameters (FPG and insulin), and most hormones. The point scale, which was the basis for the diagnosis of MetS, was also not disparate in the two groups. However, significant differences exist in lipid profile (higher LDL and triglycerides, lower HDL), body composition (higher VF), and specific hormones (lower testosterone, DHEA-S, cortisol, and higher FSH in patients with TS).

The prevalence of systolic and diastolic hypertension in our cohort was 9.1% and 11.4%, respectively (Table 3). No differences were observed between SBP and DBP among those with a 45X0 karyotype compared to those with a mosaic karyotype. Also, the incidence of IFG, abnormal lipid fraction concentrations, and, consequently, the incidence of MetS diagnosis, showed no significant differences among the participants in our study and between subgroups. Our results are consistent with the literature data of studies that assessed the risk of these disorders among individuals with Turner syndrome, treated or untreated, with rhGH and sex hormones.

The results (presented in Table 4) highlight a significant and statistically validated association between increased VF and TS. This association underscores the potential importance of VF as a differentiating factor between individuals with TS and their height- and weight-matched counterparts. Notably, no other variables, including hormonal, metabolic factors, BMI, or waist circumference (WC), demonstrated significant associations in this study.

The mean muscle strength in patients with TS was not significantly different from those in the control group. In both groups, mean muscle strength was slightly lower in those diagnosed with MetS; however, the differences were not statistically significant (Figure 1).

## 4. Discussion

The present study was designed to shed light on the metabolic and hormonal characteristics of individuals with TS who underwent effective rhGH treatment as children and adolescents and are currently still receiving sequential estrogen–progesterone therapy. An important point to emphasize is that the subjects in the study included a well-defined population of females with TS who were in their third decade of life and had reached a final height of at least 150 cm.

Demographic and anthropometric data showed no statistically significant differences between the patients with TS and the control group, suggesting that baseline characteristics, including age, BMI, height and weight were not relevant for further consideration. It is also worth noting that waist circumference between the two groups was not statistically different. This similarity in baseline characteristics reinforces the reliability of subsequent analyses comparing the two groups.

Clinical and laboratory characteristics revealed some interesting findings. Many of the studies published to date on metabolic complications in TS, including more recent ones, have often looked at patients with a lower final height, those who either did not receive rhGH, or received therapy that was not effective [19,23,24].

In our patients, BP, both systolic and diastolic, was not significantly different between patients with TS and controls. In earlier studies, especially those conducted on the pediatric population, the prevalence of hypertension in girls with TS was estimated to range from 7% [18] to 40% [19]. These differences can be partly attributed to the technique used to measure blood pressure. Studies that used 24 h ambulatory BP monitoring typically showed a higher incidence of BP compared with studies using conventional BP measurements. The effect of rhGH therapy on BP has been inconclusive, with some studies showing no effect of treatment, while a study by Pirgon et al. proved a DBP-lowering effect in rhGH-treated subjects with TS [25,26,27].

In our patient cohort, the levels of FPG and insulin exhibited similarity between the two study groups. Notably, we observed a prevalence of impaired fasting glucose (IFG) at 7.3%, a figure consistent with the reported rates in the literature for adolescents [28]. The HOMA-IR values spanned from 0.9 to 9.1, indicating a broad spectrum of insulin secretion. The findings align with Alvarez-Nava et al.’s conclusion that higher HOMA-IR values, coupled with lower insulin secretion in patients with TS, may signify a deterioration of pancreatic β-cell function [29,30]. This mechanism could potentially contribute to the early onset of type 2 diabetes mellitus (DM) in individuals with TS.

Unfortunately, our results in this regard may be biased, because we did not evaluate patients for IGT in our study. In the available literature, there are studies proving that rhGH therapy has no effect on insulin sensitivity or the secretory capacity of pancreatic b- cells [31,32]. In addition, the effect of estrogen therapy on glucose metabolism, according to many authors, was neutral, regardless of the route of estrogen administration [33,34].

Previous studies have suggested that about 50% of women with TS over the age of 21 have hypercholesterolemia, and a positive correlation exists between TC and LDL-c levels and age [20,21]. Low HDL-c levels are present in about 25% of adult women with TS [22]. Higher TC, TG and HDL-c have been described in pediatric TS cohorts [35,36]. The effect of estrogen therapy on lipid profile has been neutral in various studies, regardless of the route of estrogen administration [32,33]. Among our patients with TS, abnormalities in one or more components of the lipid profile were found in 53% of subjects. We observed significant differences in lipid profiles, with those with TS showing higher levels of low-density lipoprotein (LDL) and triglycerides (TG), and lower levels of high-density lipoprotein (HDL) compared to controls. These findings may indicate a different metabolic profile in females with TS, characterized by a more atherogenic lipid profile, which may contribute to increased cardiovascular risk.

Earlier studies suggested that about 50% of people with TS in their third decade of life and older have hypercholesterolemia, and that there is a positive correlation between TC and LDL levels and age [20,21]. In contrast, low HDL levels were present in about 25% of adult women with TS [22]. In pediatric TS cohorts, higher levels of TC, TG and HDL have been presented [35,36]. The effect of ERT on lipid profile was neutral in various studies, regardless of the route of estrogen administration [32,33]. Within our group of individuals with TS, 53% exhibited abnormalities in at least one component of the lipid profile. Noteworthy distinctions in lipid profiles were noted, as individuals with TS displayed elevated levels of LDL and TG, along with lower levels of HDL compared to the control group. These observations suggest a distinct metabolic profile in those with TS, marked by a more atherogenic lipid profile, potentially contributing to an elevated cardiovascular risk.

Furthermore, our examination of body composition in individuals with TS revealed higher VF compared to the control group. These findings align with prior studies, underscoring the altered body composition in patients with TS, potentially heightening their susceptibility to metabolic complications. The lasting effects of rhGH often used in high doses [15,16,17] or hormone replacement therapy (HRT) can significantly affect body composition. Wooten et al.’s study reported that subjects with TS treated with rhGH exhibited lower subcutaneous fat and VF compared to untreated individuals [37]. The study’s authors emphasized the need for long-term investigations to determine whether this effect persists after discontinuing rhGH therapy. Abdominal obesity, strongly linked to insulin resistance and MetS features, is notably prevalent in adult women with TS, with greater fat mass assessed through dual-energy X-ray absorptiometry (DXA) or magnetic resonance imaging (MRI) [38,39]. In contrast, adolescents with TS did not show significant differences in total body weight and VF compared to the control group [13]. This suggests that in women with TS abdominal obesity may emerge as a crucial risk factor for metabolic disorders, particularly with advancing age.

When we proceeded to analyze hormonal parameters in our study, we found that subjects with TS showed lower levels of testosterone and DHEA-S, which is often described in gonadal dysgenesis [40,41]. Moreover, we observed lower levels of cortisol in the study group, which could reflect altered hormonal regulation and a lower setup in the secretion of this hormone in patients with TS. In addition, these women had higher FSH levels, as would be expected due to gonadal dysgenesis and ovarian failure.

There are available studies that attempt to examine the relationship between cardiometabolic risk in relation to muscle strength among adults in different age groups. Although most of the studies have analyzed the population of older adults, there are also available studies among younger people, including from a gender-by-gender analysis. For example, Lee et al. showed that cardiometabolic risk was 54% lower for those with relative grip strength (absolute grip strength in the 75th percentile versus the 25th percentile) [42]. In addition, Duchowny et al. established cutoff points for muscle weakness assessed as grip strength in a nationally representative sample by race and gender among older American adults [43]. Senechal et al. determined low muscle strength for predicting MetS using a composite score of normalized strength from chest press and leg press [44]. Among middle-aged adults, Strand et al. showed an association between handgrip strength and all-cause and cardiovascular mortality, confirming the role of handgrip strength as a predictor of mortality [45]. Finally, in relation to young adults, Wilkerson et al. studied 62 collegiate football players (mean age 19.9 years) and found female leg muscle strength was associated with an increased likelihood of MetS [46]. The Gracia-Hermoso et al. study categorized muscle strength by gender and identified three categories of cardiometabolic risk (categorized as “high”, “intermediate” and “low”). In this study, among the strongest women, only 2.4% had the MetS phenotype, compared to 7.9% among women with intermediate muscle strength and 18.0% among those with low muscle strength [47]. In our study, mean muscle strength was not statistically significantly different among individuals with Turner syndrome, as well as those in the CG burdened with metabolic complications compared to those without MetS. Nevertheless, individuals with metabolic complications tended to have lower muscle strength compared to those without these complications in both the study and control groups. However, our data were based on a small number of probands and would require replication on a larger cohort.

A key finding of our study was the result of logistic regression analysis, where we confirmed a significant association between increased visceral fat and those with Turner syndrome. Of all metabolic and hormonal parameters, only the amount of VF was a significantly differentiating factor between healthy subjects and patients with TS. VF accumulation is a known cause of MetS and increased cardiovascular risk, highlighting the importance of this factor in the context of atherogenic risk assessment in subjects with TS [48,49].

The strengths of our cross-sectional study comparing 44 individuals with Turner syndrome with an age-matched control group is that the research question is clearly posed. Comprehensive data collection, including anthropometric, metabolic and endocrine measurements, increases the depth of the study. The use of logistic regression provides robust statistical analyzes revealing the association between increased visceral fat and Turner syndrome.

However, there are limitations to consider. Initially, the foundational data concerning anthropometry and therapy was retrospectively gathered from patient medical records. Given that anthropometry was conducted by various examiners, the possibility of observer variation exists. Nevertheless, our treatment protocols for growth hormone (GH) and puberty induction are standardized within the institution, reducing the susceptibility to such variations. Secondly, due to the cross-sectional design of the study, the investigation into the development of metabolic comorbidities into adulthood was not undertaken. To substantiate the findings, further research with larger sample sizes and longitudinal designs is imperative. Thirdly, the TS group includes individuals with mosaic karyotypes, which may exhibit milder metabolic abnormalities in comparison to classic karyotypes.

## 5. Conclusions

In light of effective rhGH treatment in the past, enabling the attainment of a height of at least 150 cm, it is essential to recognize that female patients with TS face an elevated likelihood of developing components of MetS and experiencing a significant increase in visceral fat (VF). Our findings emphasize the importance of regular metabolic screenings for individuals with TS, irrespective of factors such as age, height, weight, and genetic karyotypes (including 45X0 and mosaic karyotypes) and suggest the potential benefits of lifestyle modification, building more muscle strength and weight control strategies. It is crucial to note that, unlike straightforward growth hormone deficiency (GHD) groups, patients with TS may not derive as much benefit from rhGH therapy.

## Figures and Tables

**Figure 1 biomedicines-12-01034-f001:**
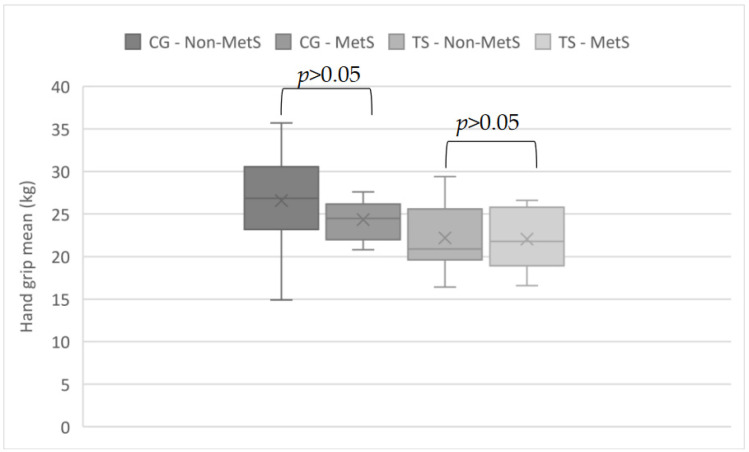
The comparisons of Handgrip mean (kg) in patients with Turner syndrome (TS) with and without metabolic syndrome (MetS) and controls with and without MetS.

**Table 1 biomedicines-12-01034-t001:** Demographic and anthropometric characteristics.

Variables	TS (*n* = 44)	CG (*n* = 52)	*p*
	Mean ± SD or Median (Interquartile Range)		
age [years]	23.7 ± 4.01	23.1 ± 5.73	0.789
BMI [kg/cm^2^]	27.3 ± 3.11	26.4 ± 4.21	0.735
height [cm]	158 (156.3–161.6)	159 (156.7–161.8)	0.869
weight [kg]	62.9 ± 4.61	63.5 ± 3.06	0.751
WC [cm]	77.40 ± 2.25	76.80 ± 2.7	0.197
Handgrip (kg)	22.2 ± 4.30	26.1 ± 5.3	0.232
NGS	0.39 ± 0.09	0.41 ± 0.10	0.351

BMI—body mass index; WC—waist circumference; NGS—measured as handgrip strength (kg)/body mass (kg); TS—patients with Turner syndrome; CG—control group.

**Table 2 biomedicines-12-01034-t002:** Clinical and laboratory characteristics.

Variables	TS; SD (*n* = 44)	CG; SD (*n* = 52)	*p*
SBP [mm Hg]	119.03 ± 18.14	116.33 ± 13.94 (52)	0.192
DBP [mm Hg]	76.42 ± 10.35	74.04 ± 11.99 (52)	0.210
FPG [mg/dL]	87.9; 8.44 (44)	89.6; 6.75 (52)	0.297
Insulin [mU/mL]	12.2; 5.09 (44)	9.4; 6.22 (52)	0.102
HOMA IR	2.78; 1.15 (44)	2.67; 1.62 (52)	0.218
Vitamin D [ng/mL]	29.3; 5.63 (42)	26.8; 7.5 (50)	0.310
TC [mg/dL]	181; 54.6 (42)	171; 27 (52)	0.261
LDL [mg/dL]	133; 23.5 (42)	106; 21.8 (52)	<0.01
HDL [mg/dL]	53.4; 13.2 (42)	72; 19.3 (52)	<0.01
TG [mg/dL]	116; 48.3 (42)	82; 32.1 (52)	<0.01
Ca total [mmol/L]	2.35; 0.19 (41)	2.43; 0.08 (49)	0.067
Ca ionized [mmol/L]	1.27; 0.03 (32)	1.27; 0.04 (47)	0.348
TSH [μIU/mL]	2.85; 1.55 (41)	2.38; 1.85 (51)	0.065
FT4 [pg/mL]	1.30; 0.241 (41)	1.24; 0.26 (51)	0.119
FSH [mIU/mL]	48.9; 28.1 (41)	6.00; 1.89 (50)	<0.01
Estradiol [pg/mL]	45.1; 29.6 (40)	38.7; 28.6 (50)	0.863
Prolactin [ng/mL]	21.1; 15.2 (38)	18.9; 11.1 (50)	0.357
Testosterone [ng/mL]	0.267; 0.224 (38)	0.361; 0.159 (45)	0.032
DHEA-S [μg/mL]	159; 76.3 (41)	248; 130 (46)	<0.01
Cortisol [μg/mL]	13.8; 5.37 (40)	18.9; 7.18 (49)	<0.01
ACTH [pmol/L]	14.8; 7.11 (40)	16.8; 6.8 (50)	0.064
IGF-1 [μg/L]	154; 75 (40)	166; 87 (49)	0.462
BF [kg]	22.9; 12.67 (44)	20.8; 9.85 (52)	0.619
VF [kg]	0.61; 0.47 (44)	0.39; 0.22 (52)	<0.01

Abbreviations: SBP—systolic blood pressure; DBP—diastolic blood pressure; FPG—fasting plasma glucose; HOMA-IR—homeostatic model assessment for insulin resistance; TC—total cholesterol; LDL—low-density lipoprotein; HDL—high-density lipoprotein; TG—trigliceride; Ca—calcium; TSH—thyroid-stimulating hormone; FT4—free thyroxine; FSH—follicle-stimulating hormone; DHEA-S—dehydroepiandrosterone sulfate; ACTH—Adrenocorticotropic hormone; IGF-1—inulin-like growth factor 1; BF—body fat; VF—visceral fat; TS—patients with Turner syndrome; CG—control group.

**Table 3 biomedicines-12-01034-t003:** Prevalence of hypertension and metabolic disorders in females with TS among participants in our study compared with literature data from studies of individuals with TS, both treated and untreated, with rhGH and sex hormones. The members of our study, additionally, were divided into groups according to karyotypes: monosomy 45X0 vs. other mosaic karyotypes. The table is compiled from different sources [18,19,20,21,22].

Metabolic Parameters	Total Affected *n* = 44 (%)	45X0 *n* = 21 (%)	Mosaics *n* = 23 (%)	*p* 45X0 vs. Mosaics	Literature [18,19,20,21,22]
Systolic hypertension	4 (9.1)	2 (9.5)	2 (8.7)	ns	7–40%
Diastolic hypertension	5 (11.4)	3 (14.3)	2 (8.7)	ns	7–40%
IFG	3 (6.8)	1 (4.7)	2 (8.7)	ns	7–8.3%
High TC (>95th centile)	6 (13.6)	3 (14.3)	3 (13.0)	ns	20.5–37.5%
High LDL (>95th centile)	4 (9.1)	2 (9.5)	2 (8.7)	ns	8–26%
High TG (>95th centile)	6 (13.6)	3 (14.3)	3 (13.0)	ns	7.7–35%
Low HDL (<10th centile)	5 (11.4)	2 (9.5)	3 (13.0)	ns	7.9–18.4%
High FPG	4 (9.1)	2 (9.5)	2 (8.7)	ns	7.2–32.1%
MetS	7 (15.9)	4 (19.0)	3 (13.0)	ns	16.2–34.3%

Abbreviations: IFG—impaired fasting glucose; TC—total cholesterol; LDL—low-density lipoprotein; HDL—high-density lipoprotein; TG—trigliceride; FPG—fasting plasma glucose; MetS—metabolic syndrome; TS—patients with Turner syndrome; ns—not significant.

**Table 4 biomedicines-12-01034-t004:** Main results of the multiple logistic regression analysis for factors related to metabolic parameters and hormonal levels as well as body composition, with individuals with TS or controls as the dependent variable.

Variable	Coefficient	Std. Error	z-Value	*p*-Value	Beta
VF	0.02299	0.01122	2.791	0.042	1.333

## Data Availability

Data are available on special request after contacting author (M.K.).

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
