# Peer review of "Are Young People with Turner Syndrome Who Have Undergone Treatment with Growth and Sex Hormones at Higher Risk of Metabolic Syndrome and Its Complications?"

_biomedicines, 2024, doi:10.3390/biomedicines12051034_

Round 1

Reviewer 1 Report

Comments and Suggestions for Authors

The manuscript “Are young people with Turner syndrome who have undergone treatment with growth and sex hormons at higher risk for metabolic syndrome and its complications?” describes a clinical investigation performed on people with Turner syndrome and a matched control group of women to compare risk factors of metabolic syndrome, blood lipid profiles, levels of some hormones and body composition. The objective of the study was to assess the prevalence of the metabolic syndrome in young women with Turner syndrome with the goal to introduce more efficient management strategies. The study is carefully planned and performed and the text well written. The references are pertinent and the statistical analysis correctly applied. However, there are some concerns with the manuscript:

Major concerns:

1: The introduction states that: “This study is based on the hypothesis that effective hormone therapies may influence the prevalence of MetS parameters, specifically focusing on their impact on diabetes mellitus, in individuals with TS. The primary objective of this investigation is to assess the prevalence of MetS elements, with a particular emphasis on diabetes mellitus, in young women with TS.”

From this text, we are to understand that the focus is diabetes type 2. However, there were no differences between the TS group and the control group. Instead, the interesting parameters were blood lipids, FSH and cortisol. Please rephrase the introduction to correctly introduce the clinical findings.

2: The authors state that they have “the goal of guiding more effective clinical management strategies.” If this is a goal, please elaborate on how TS should be managed.

3: The authors explain that: “The exclusion criteria for the study were a height of less than 150 cm or more than 162 cm. It aimed to minimize potential confounding factors related to height variations.”

Please explain the rationale behind this and specify what those confounding factors might be. This is important to allow the reader to evaluate whether this exclusion criterium was valid.

Minor concerns:

1: Line 37: “Lipid profiles showed higher levels of LDL and TG and lower HDL in subjects with TS, indicating an atherogenic profile. Body composition analysis showed increased visceral adipose tissue in those with TS. Hormonally, individuals with TS showed lower levels of testosterone, DHEA-S and cortisol, as well as elevated FSH. Metabolic abnormalities were common in individuals with TS, including hypertension, impaired fasting glucose levels and unfavorable lipid profiles”.

The unfavorable lipid profile was described in line 37 and there is no need to repeat that fact. Please rewrite to avoid repetition.

2: Line 100: In conclusion, not “in concluding”.

3: Diagnosis of metabolic syndrome: It is confusing that you give the criteria also for men. No men participated in the study.

4: Line 166: Please explain the acronym MAP.

Comments on the Quality of English Language

The English language is overall very good, with some minor editing required.

Author Response

Dear Reviewer,

Thank you very much for your comments. They certainly improved the quality of our article. We have made every effort to answer them. Below there is the answer to your comment.

Major concerns:

  1. Indeed, this statement was misleading. We have rephrased it.
  2. According to your suggestion we have modified the aim of the study.
  3. Indeed, we agree, we greatly appreciate your astute observation. We have edited the purpose of the paper, explaining why body height was important in the analysis of metabolic factors and body composition.

Minor concerns:

  1. It has been changed according to yours suggestions
  2. It has been changed
  3. The diagnostic criteria of metabolic syndrome for men have been removed as you had suggested.
  4. The acronym MAP has been explained.

Additionally, we have revised the article and made some language editing.

We hope that you will find the revisions appropriate and allow the manuscript to be published.

Reviewer 2 Report

Comments and Suggestions for Authors

The article entitled: Are young people with Turner syndrome who have undergone

treatment with growth and sex hormones at higher risk for metabolic syndrome and its complications? Is really important in the post-pandemic era. During the COVID-19 time, an increase in psychological problems has been observed. Moreover, psychological disorders correlate well with metabolic problems. In light of the above the genetic abnormalities like Turner syndrome were relegated to the shadow area. Therefore this manuscript has been found by me as interesting for the broad audience not only scientific. The presented results disclose the low level of unsaturated fat acids in the case of patients with TS which correlated positively with a high level of visceral adipose tissue. Therefore again has been shown that obesity leads to different health problems ( e.g.: cardiovascular ).

The used methodology in this study is correct and well-described. Moreover, the article is well-written and readable. The references are correctly selected.

Please add the single paragraph abstract.

Based on the above I can recommend the article for publication.

Author Response

Dear Reviewer,

Thank you very much for your comments.

We have made some minor modifications according to suggestions of another reviewer. 

We hope that you will find the revisions appropriate and allow the manuscript to be published.